# Non-Named Entities – The Silent Majority

Pierre-Henri Paris, Fabian Suchanek

Télécom Paris, Institut Polytechnique de Paris, France
`first.last@telecom-paris.fr`

**Abstract.** Knowledge Bases (KBs) usually contain named entities. However, the majority of entities in natural language text are not named. In this position paper, we first study the nature of these entities. Then we explain how they could be represented in KBs. Finally, we discuss open challenges for adding non-named entities systematically to KBs.

## 1   Introduction

RDFS Knowledge Bases (KBs) such as DBpedia, YAGO, and Wikidata usually contain *named entities* – e.g., people, locations, and organizations. These entities are canonicalized, they are equipped with facts, and they belong to classes that are arranged in a taxonomy. The facts of a KB can be extracted, e.g., from natural language text. However, the majority of entities in real-world natural language text are not named. Consider for example the sentence "The Arab Spring resulted in a contentious battle between a consolidation of power by religious elites and the growing support for democracy" (taken from a Wikipedia article). This sentence is clearly factual, and thus interesting for information extraction. However, it contains only a single named entity. Many fact-extraction techniques will simply not consider the other entities. Thus, these entities remain "silent", and the statement stands no chance of making it into an RDFS KB.

The problem of extracting facts from such sentences is, of course, as old as information extraction itself. In this position paper, we look at the problem from a Semantic Web point of view. Technically, we look at *noun phrases*, i.e., sequences of words that take the grammatical function of a noun. In English, noun phrases can be the subject or object of a sentence, they fall under the *NP* node in phrase structure grammars, and they can contain determiners, adjectives (possibly modified by adverbs), one or several nouns (the last of which is the *head noun*), prepositional phrases (such as "in the Arab World"), relative clauses (introduced, e.g., by "which"), and other modifiers of the head noun. All noun phrases refer to *entities*. Some noun phrases are proper names (capitalized in English), and thus refer to *named entities* – but the vast majority are not. In a slight misuse of terms, and in line with [1,5], we call these noun phrases *non-named entities*.

In this article, we first study the nature of non-named entities manually on Wikipedia articles (Section 2). Then we discuss how they can be extracted and added to RDFS KBs (Section 3). Finally, we discuss what would have to be done to make non-named entities first-class citizens in RDFS KBs (Section 4).

## 2   Non-Named Entities

Several works have investigated the nature of non-named entities, albeit only as a side-product: The work of [5] studied Named Entity Recognition (NER) for resource-poor languages, and found that 86% of noun phrases in the studied corpus were non-named entities. The work of [1] studied the performance of NER models on different corpora, and manually annotated noun-phrases, classifying them into ACE categories. Due to the choice of literary texts, 67% of phrases refer to people.

To further investigate the nature of non-named entities, we conducted a manual analysis of noun-phrases in Wikipedia articles. Our choice is motivated by the fact that Wikipedia is a widely used standard reference, which also feeds KBs such as DBpedia and YAGO. We focus on the "featured articles", and choose one article from each of the 30 topics[1]. We automatically extract (and manually verify) noun phrases from the abstract of the article. We consider noun phrases that are sequences of nouns, adjectives, adverbs, prepositions, and determiners – 1924 in total.

We find that 78% of noun phrase heads are non-named entities. Of these, 63% are singular, and thus refer to a single entity ("a French book"). The others are plural, and could thus be understood as ad-hoc concepts ("French books"). Inspired by the YAGO 4 taxonomy [9], we annotated the noun phrases by the top-level classes of schema.org combined with the top-level classes of BioSchema.org. Unsurprisingly, a large part of the non-named entities are Intangibles – such as "support", "gain", or "operation". Creative works (such as "song") and people ("singer") are also frequent. 32% of non-named entities are determined ("the man"), which makes it more likely that they are central to the text. At 19%, mass nouns ("fame") are also rather frequent, while quantified expressions ("3 men") are not frequent. We counted as anaphoras any reference (by pronoun or determined noun phrase) to a preceding entity. They are rather infrequent. Some entities are qualified by an anaphora ("one of his siblings"). While one third of the noun phrases are unaccompanied, 38% are modified by an adjective, and 34% have a preposition. Inter-annotator agreement was 88%.

### Head of Noun phrases

| | | |
|---|---|---|
| Named | 428 | 22% |
| Non-named | 1496 | 78% |
| Total | 1924 | 100% |

### Non-named: Plurality

| | | |
|---|---|---|
| singular | 937 | 63% |
| plural | 590 | 37% |

### Non-named: Class

| | | |
|---|---|---|
| action | 124 | 8% |
| product | 67 | 4% |
| person | 147 | 10% |
| taxon | 40 | 3% |
| event | 168 | 11% |
| intangible | 418 | 28% |
| place | 121 | 8% |
| organization | 75 | 5% |
| medicalentity | 3 | 0% |
| creativework | 310 | 21% |
| biochementity | 23 | 2% |

### Non-named: Nature and Modifiers

| | | |
|---|---|---|
| undetermined | 590 | 39% |
| determined | 485 | 32% |
| quantified | 109 | 7% |
| anaphora | 92 | 6% |
| qualified by an. | 133 | 9% |
| mass noun | 288 | 19% |
| preposition | 503 | 34% |
| contains named | 167 | 11% |
| adjective | 571 | 38% |
| adjacent nouns | 188 | 13% |
| no modifiers | 506 | 34% |

---

[1] https://en.wikipedia.org/wiki/Wikipedia:Featured_articles

## 3   Adding Non-named Entities to RDFS KBs

**Extracting non-named entities.** Semantic parsers (such as FRED, K-Parser, Pikes, or Graphene), Abstract Meaning Representation systems, and structured discourse representation systems (such as [2]) build a semantic structure on top of a dependency parse of the input sentence. In this representation, non-named entities can figure just like named entities. The named entities can be mapped to existing entities in a KB, and the others remain unmapped. Some of these systems also perform co-reference resolution, so that anaphoras are resolved, and the semantic structure becomes a graph (as opposed to a tree). Open Information Extraction (IE) systems (such as OpenIE, StuffIE, NestIE, MinIE, or Claus-IE) build triples of subject, predicate, and object, all of which can be arbitrary phrases (instead of canonicalized entities). Both representations can thus contain non-named entities (see, e.g, [4] or [6] for details about these systems).

**Modeling non-named entities.** The question is now how these non-named entities can be modeled in an RDFS KB. One possibility is by the means of classes: Plural phrases can become classes, and singular phrases can become anonymous instances of ad-hoc classes (so that two occurrences of "a French book" can be two instances of that class). These classes could then become subclasses of the superclass determined by the head noun ("French book" $\rightarrow$ "book"), or else directly the top-level class ("CreativeWork") – which could be determined by multi-class classification. Anaphoras (such as pronouns or determined nouns) would have to be replaced by their referents. Noun phrases that contain nested noun phrases would have to receive special treatment ("the gas in the balloon", e.g., would have to be linked to "the balloon"). Numbered noun phrases could be replicated appropriately. Mass nouns would have to become classes: Even if they do not allow a plural, they can still have instances (as in "the fame that Elvis achieved").

**The Gap.** Even if some non-named entities can be added straightforwardly to RDFS KBs, there is still a gap between the noun phrases in a text and the entities in an RDFS KB. There is currently no way that an RDFS KB can integrate a non-named entity such as "the growing support for democracy in many Muslim-majority states". Such an entity appears to combine several entities, it would be difficult to attach to a class other than "Intangible", it could semantically overlap in unspecified ways with other entities (such as "the support for democracy in Muslim states"), and it is difficult to equip with facts, let alone axioms. Let us now detail these challenges.

## 4   Bridging the Gap

**Knowledge Representation.** The modeling of non-named entities depends on the modeling of classes and instances in the target KB. Currently, some KBs (such as ConceptNet, BabelNet, Quasimodo, and in general common sense KBs [7]) contain mainly classes. Others (such as WebIsALod, WebOfConcepts, NELL) make no difference between instances and classes. Again other KBs mix

classes and instances (Wikidata makes "human" both an instance and a class). Some KBs duplicate classes and instances (DBpedia, e.g., contains a class and an instance called "book"), and others keep them separate (e.g., YAGO). These models may need to be reconsidered when non-named entities are added to the KB (see, e.g., [3] or [10] for surveys about such KBs).

Another challenge are plural phrases (such as "scientists", "many MLB batting records", or "hundreds of soldiers"), which generate an unspecified number of instances. These would potentially have to be modeled by axioms. OWL could provide the necessary semantics here. The same is true for all-encompassing noun phrases such as "all other non-ferrous metals". Some non-named entities would better be modeled as relations ("the knowledge of irrigation", "his invention of an early instant coffee process", or "the focus of their individual chapters"). Vagueness is another major challenge ("large-scale settlement" or "various notions"), as are noun phrases that reify statements ("the perception of hieroglyphs as purely odeographic") [8].

**Canonicalization.** Entities would have to be canonicalized, so that synonymous noun phrases are merged into one non-named entity ("the rise of the stock market" and "the surge of the stock market"), but distinct entities are kept apart (rises of different stock markets). This harbors challenges in the dimension of time (two different rises of the same stock market) and more generally in determining whether similar non-named entities in different contexts are the same (two stock market rises mentioned in different texts).

**Facts.** Non-named entities can require elaborate statements about classes (think of "pacific winds", "dormant volcano", or "his characteristic surrealist style"). A particular challenge are phrases that make sense only in connection to other phrases, especially comparatives, superlatives, and temporal comparisons ("the first woman to pilot her own baloon", "more powerful centers appearing to the south", "a growing hostility toward factual discussion", etc.). All of these may require elaborate axioms. Currently, instance-oriented KBs tend to make crisp boolean statements ("$X$ was born in $Y$"), while class-oriented KBs tend to make weak statements ("Elephants are gray" means that elephants are typically gray, not necessarily all of them). To model non-named entities, one would probably need stricter relationships for instances and laxer ones for classes. This cohabitation, however, has not yet been studied.

## 5    Conclusion

In this position paper, we have argued that non-named entities make up a large majority of noun phrases in natural language text. We have analyzed their nature in a manual study of Wikipedia articles, and we have discussed how such noun phrases could be extracted and added to KBs. Finally, we have listed a number of challenges that still remain, indicating that we are still a long way from making full use of natural language text for RDFS KBs. All our data is available at https://phparis.net/posts/non-named_entities.

**Acknowledgements.** This work was partially funded by the ANR IA grant ANR-20-CHIA-0012 ("NoRDF").

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
