# OpenReview forum: "Non-named entities - the silent majority"
_eswc-conferences.org/ESWC/2021/Conference/Poster_and_Demo_Track — ESWC2021 P&D_

### Official Review · AnonReviewer4 · 2021-04-06
**Interesting discussion**

**Rating:** 7
**Confidence:** 4

**Review:**

This position paper discusses non-named entities in information extraction and knowledge representation.

In general, the topic is very interesting. The authors analyzed 30 Wikipedia articles and drew a few useful conclusions.

Still, I have the following concerns.

(1) I think some non-named entities (e.g., "the man") can be solved by entity coreference resolution. They may not be a real issue here.

(2) I am thinking about whether it is necessary to extract everything from text. For example, given "the growing support for democracy in many Muslim-majority states", is it really necessary to extract every entity and fact from such a sentence? How can we exploit the extracted knowledge?

(3) I would be better if the discussion could address OWL rather than RDFS. Some representation issues may be solved using OWL.

Please include hyperlinks in the text (e.g., as footnotes).

**Anonymity:**

Yes, I would like my review to remain anonymous.

---

### Official Review · AnonReviewer3 · 2021-04-14
**This position paper discusses the problem of linking non-named entities into KBs.**

**Rating:** 8
**Confidence:** 4

**Review:**

This is a position paper that discusses a very interesting topic. In fact, I have also noticed this problem recently and I believe it is a very important problem for practical entity linking and typing.

**Anonymity:**

Yes, I would like my review to remain anonymous.

---

### Official Review · AnonReviewer2 · 2021-04-15
**Interesting position paper, but the position is not clear**

**Rating:** 6
**Confidence:** 4

**Review:**

This paper was a short but interesting and engaging read. However, at the end of the read I had the feeling that the paper fell a bit short of its objectives.

First of all, the fact that non-named entities are the majority of entities is purported as a debated topic, but in my view this is a sort of common knowledge. It is quite evident that named entities represent a minority of total entities. Something that is just suggested but not investigated further is the fact that KBs tend to contain way more named than non-named entities; so there is a gap between what KBs can represent and what is in the texts. It could be also interesting to understand if this happens because the meaning of named entities is easier to understand and define, or because we can produce NER systems that are very accurate but detecting (non named) entities is way less accurate (because entities depend on the task, the KB, etc...).

Another problem (regarding the "position") is that it is not clear if the authors believe in the challenges to be solvable or not. In particular, when I read that "synonymous noun phrases are merged into one non-named entity", this is extremely ambitious in my opinion, especially if we are talking about entities that are described by rather long phrases. We already have sometimes problems in mapping short fragments of text to the same entity; taking into account the richness of human language, there could be many variations of expressing a concept like "the support for democracy in Muslim states", and not all of them could be exact synonymous (for instance, if we replace Muslim with Arab we are excluding places like Indonesia).

It seems also that the solution of some issues (such as the choice of modeling some entities with relations) really depends on the use that one wants to make of the entities, so at the end it would be a subjective decision.

In conclusion, the paper is very well written, and I appreciate the great work done on the analysis aspect and the identification of challenges. What it is still missing is probably a more clear position regarding these challenges.

**Anonymity:**

Yes, I would like my review to remain anonymous.

---

### Decision · Program_Chairs · 2021-04-19

Accept